# HOT CRT—The Effective Combination of Conventional Cardiac Resynchronization and His Bundle Pacing

**DOI:** 10.3390/medicina58121828

**Published:** 2022-12-12

**Authors:** Peter-Stephan Wolff, Anna Winnicka, Adam Ciesielski, Malte Unkell, Grzegorz Zawadzki, Agnieszka Sławuta, Jacek Gajek

**Affiliations:** 1Department of Cardiology, Augusta Hospital Düsseldorf, Academic Teaching Hospital of the University, Faculty of Health, 40472 Düsseldorf, Germany; 2Department of Cardiology, Multidisciplinary Public Hospital, 67-100 Nowa Sól, Poland; 3Students’ Scientific Association, Department of Emergency Medical Service, Wroclaw Medical University, 51-616 Wroclaw, Poland; 4Department of Internal and Occupational Diseases, Hypertension and Clinical Oncology, Wroclaw Medical University, 50-566 Wroclaw, Poland; 5Department of Emergency Medical Service, Wroclaw Medical University, 51-616 Wroclaw, Poland

**Keywords:** CRT, His-bundle pacing, HOT-CRT, AF, IVCD, Non-LBBB

## Abstract

*Background and Objectives*: Cardiac Resynchronization Therapy (CRT) has, besides its benefits, various limitations. For instance, atrial fibrillation (AF) has a huge impact on the therapy efficacy. It usually reduces the overall BiV pacing percentage and leads, inevitably, to lack of fusion beats. In many patients with heart failure that could benefit from resynchronization, the QRS morphology is often IVCD and atypical, or non-LBBB, which further diminishes the CRT response. In those cases, we established His pacing combined with LV pacing as a feasible option to reduce the impact of AF on the CRT response and regain partially physiological ventricular activation to improve the electromechanical sequence. *Materials and Methods*: We implanted two patients with AF, HF, EF < 35%, NYHA II-III and QRS > 150 ms with CRT-D systems modified to HOT-CRT and observed their clinical, ECG and echocardiographic improvements over a follow-up period of three months. *Results:* In both patients we observed improvements of the initial parameters. We were able to shorten the QRS duration to approx. 120 ms, improve NYHA functional class, increase the EF by approximately 12% and distinctly reduce mitral regurgitation. *Conclusion:* Since the conventional CRT reaches its limits within this specific patient group, we need to consider alternative pacing sites and the effective combination of them. Our results and respectively other studies that are also mentioned in the current guidelines, support the feasibility of HOT-CRT in the above mentioned patient group.

## 1. Introduction:

Cardiac resynchronization therapy (CRT), is the standard procedure for all patients with cardiomyopathy, left bundle branch block (LBBB) and advanced heart failure (HF) [1]. With this conventional procedure, however, up to one third of CRT patients are non-responders. This specific group of patients have various underlying conduction pathologies, which cannot or, at least not efficiently, be corrected by the standard CRT system alone [2]. Even less efficacy can be achieved in non-typical LBBB morphology. In an exemplary small study, the efficacy of CRT was compared between patients having LBBB, right bundle branch block (RBBB) and non-specific intraventricular conduction delay (IVCD). Patients with LBBB had, by far, the best therapy response, while the prognosis declined considerably for those with RBBB and very pronounced in IVCD [3]. In particular, the last result is most probable the effect of the underlying disease. The non-classic LBBB morphology is often the result of ischemic myocardial changes, which delay the intraventricular conduction in an unpredictable way, related to myocardial necrosis and fibrotic scarring. In many CRT recommendations, the authors are trying overcome this issue by establishing a longer QRS duration as indication, but this does not provide much help as the results in those patients are insufficient [4].

Even less success yields the standard resynchronization in patients with chronic atrial fibrillation. This condition demands constant right ventricle pacing for appropriate left ventricle pacing timing. In a vast majority of the patients the desired QRS complexes shortening is very modest, if at all present [5,6]. With this regard, the combination of non-specific IVCD with chronic atrial fibrillation constitutes a very demanding substrate for resynchronization. The originally developed concomitant direct His bundle pacing and left ventricular pacing (HOT-CRT after P. Vijayaraman) is a promising method to overcome the described problem [7]. 

## 2. Purpose

The aim of this case reports was to discuss the effectiveness of modified resynchronization in two patients with chronic AF, regarding the clinical electro- and echocardiographic results.

## 3. Cases Presentation

Our first patient was male (59 y.o.) with an ischemic cardiomyopathy and permanent AF. In 2010 he was implanted with and ICD-VR, had in 2016 ICD replacement and in 2019 the up-grade to HOT CRT. He had a tendency to bradycardia and ventricular rhythm of approximately 50-55/min. In the past, there were multiple VF episodes and effective defibrillation. Furthermore the echocardiography showed features of advanced mitral regurgitation (+++/++). Initially he presented with NYHA III, EF 30%, LVEDD 6.9 cm and LA 5.9 cm. The patient was implanted with a Compia QUAD CRT-D and a practically 100%, selective His bundle pacing was achieved from atrial channel. The device programming was set to His-LV only pacing with AVD of 60 ms.

Three months later, the follow up showed an improvement to NYHA II, EF 43%, LVEDD 6.3 cm, LA 5.3 cm and reduced mitral regurgitation (++/+). The threshold for HBP was initially 2.1 V assessed by 0.8 ms stimulus duration. During follow-up it did not change. It was set permanently at 3.5 V and 0.6 ms.

The ECG results of the resynchronization procedure including standard BiV (LV—20 ms) are presented in Figure 1. 

The second patient was male (60 y.o.) with an ischemic cardiomyopathy and permanent AF. In the medical record we found a MI and the consecutive PCI of LAD. In 2019 followed the implantation with cardiac resynchronization implementing HOT-CRT concept.

Initially, he presented with NYHA III, EF 21%, LVEDD 6.2 cm and LA 4.5 cm. Echocardiography showed features of advanced mitral regurgitation (+++/++). We implanted him with a Compia Quad CRT-D. The atrial port of the device was used for direct His Bundle pacing which was selective. The increase of metoprolol dosage resulted in 100% of pacing. The device was set as previously to His-LV only with AVD of 60 ms with HBP from atrial device port.

The follow up was three months later and showed improvement in NYHA II, EF 31%, LVEDD 6.2 cm, LA 4.9 cm and reduced mitral regurgitation (++/+). The threshold at the implantation was 1.6 V by 0.8 ms and during follow-up we measured 1.9 V by 0.8 ms. It was set permanently at 3.5 V and 0.6 ms stimulus duration.

The ECG results are displayed in Figure 2.

In both cases the EF assessment was based on LV volume changes, calculated by tracing the LV endocardium in 4-chamber and 2-chamber views at the end-systolic and end-diastolic phases and calculating the mean value of those two results. To achieve high percent of BiV pacing the patients were treated with metoprolol and digoxin in appropriate doses. In both cases the selective HBP was achieved, mainly to not introduce another kind of dyssynchrony in the ventricular systolic pattern.

## 4. Discussion

Conduction abnormalities, like a distal LBBB or IVCD, are challenging problems during the CRT implantation and follow-up. Even various attempts to find the best pacing site by observing the change in QRS morphology are limited to the given anatomical structure of the heart, or more accurate to the coronary sinus branches. Although the QRS narrowing does not have the most reliable prognostic value, the electrical resynchronization it reflects is quite useful in predicting CRT success [8]. Now since the release of the quadripolar LV lead, we were able to increase the pacing area, but this option is still very limited by the anatomy. 

Besides the different opinions on the CRT response and the feasible options to improve it, there is a consensus, that a high percentage of RV pacing is deleterious to a good therapy response. At this point we are confronted with another problem that cannot be solved with a conventional CRT-System. Atrial fibrillation is common in the group of heart insufficient patients. Many can be diagnosed at their first evaluation, others develop it in the process of structural heart changes of and consecutive insufficiency. Those who have an indication for a CRT-System, can be expected to have a high percentage of RV stimulation and won’t benefit from this approach.

The Figure 3 displays different scenarios, where the CRT is limited and how it could be improved by pacing another structure. Picture A and B show the few options and the realistic portion of LV pacing area, where the electrodes should be placed, to reach the furthest distance between them and thus lead to an optimal spread of the electrical activation. Picture C and D, on the contrary, represent the advantage of an additional lead in the His Bundle. During atrial fibrillation, His Bundle pacing would enable us to have a homogenous RV septum activation [9,10]. Furthermore His pacing will not change the area of LV maximum delay as classic RV pacing might do. Picture D shows the theory of gaining the optimal axis between the His bundle activated RV septum and more independently of the position of the LV lead. Besides those advantages, we get the possibility, to correct some of the rather complicated conduction abnormalities. In patients with progressive cardiomyopathy, other IVCDs can coexist in addition to the LBBB, which His Bundle pacing alone could not resolve. In this case, we believe it makes sense to optimize the CRT with both His and LV pacing as it provides the possibility of better fusion pacing less related to anatomically related suboptimal LV position. This novel pacing method was named by Vijayaraman et al. as HOT-CRT (His Optimized CRT) [7]. The authors implanted permanent HBP and an additional LV lead in 27 patients, who were referred for a CRT. 17 patients had initially LBBB, 5 IVCD and 5 right ventricular pacing. The pacemakers were programmed to allow a delay between HB- and LV pacing, similar to the intrinsic His-ventricular delay. This procedure was successful in 25 patients. The QRS duration initially was 183 ± 27 ms and was narrowed down, to 162 ± 17 ms during biventricular pacing, to 151 ± 24 ms during solely HBP, and further to 120 ± 16 ms during HOT-CRT. After 14 ± 10 months also Echocardiographic results improved, from 24 ± 7% to 38 ± 10% in EF and NYHA class reduced from III to II. Combined His and LV pacing made it possible, to overcome distal branch blocks and IVCD, which is hard to be corrected by solely RV/LV pacing.

Furthermore Boczar et al. observed an another study group of 14 patients, aged 67.4 ± 10 years, who were implanted the same way with direct HBP and LV electrode. The patients initially had a mean QRS duration of 159.2 ± 28.6 ms and LVEF was 24.36 ± 10.7% [11]. After a follow up period of 14 months, they observed that the LVEF increased from 24% to 38%, LVEDD decreased from 59 mm to 47 mm and the average QRS duration was shortened from 159 ms to 128 ms. HBP was achieved in 97% and as a result, 92.3% of them showed an improvement in NYHA class.

Interestingly, previously Boczar et al. also described a case which is directly related to the HOT-CRT technique. A female patient with dilated cardiomyopathy, NYHA III and permanent AF underwent implantation with CRT. His-pacing from the atrial and LV pacing from the LV channel resulted in a shortened QRS complex, from initially 178 ms, to 160 ms by sole HBP and further to 130 ms, by DDD pacing from HIS and LV (40 ms delay between the stimuli). LBBB pattern was completely resolved and the Echocardiograph showed abolished LV rocking, and synchronous activation of LV free wall. After 12 months, the follow up NYHA decreased from III to II and LVEF increased from 15 to 35% [12]. 

In the cohort of patients implanted with a CRT, there are different parameters to evaluate their individual therapy success. Underlying structural, hemodynamic and electrophysiological changes divide this group into different responses to CRT. Looking at studies from the last years, an estimated one third of all patients that received cardiac resynchronization are non-responders. Atrial fibrillation, even if not present during the initial implantation, can be found in one out of four patients with CRT and generally worsens their prognosis. The most important values for the CRT response, is the percentage of biventricular pacing (BiV%). Over 99% is observed to yield the highest reduction in mortality by 24%, while BiV% < 94.8% is associated with an 19% increase in mortality. AF patients tend to exhibit a loss in atrioventricular synchronicity, which leads to insufficient CRT delivery and in case of CRT-D even to inappropriate shocks [13]. Ultimately the AV dyssynchrony can lead to fusion and pseudo-fusion beats that depict the ineffective biventricular capture. A study by Ganesh S. Kamath et al. addressed this problem by observing a patient group with permanent AF who underwent CRT [14]. While the device interrogation showed >90% BiV pacing, the 12 lead Holter ECG unveiled, that only 47% received >90% of fully paced beats in 24h. The other 53% of patients had 16.4 ± 4.6% fusion and 23.5 ± 8.7% pseudo-fusion beats. Pacing counters significantly overestimated the amount of effective BiV pacing, which leads to a wrong perception of the therapy success.

Since the results, of the MADIT-CRT trial, we have some important insight on the CRT response in different kinds of LBBB morphology [15]. While a patient with typical LBBB (>150 ms) was more likely to benefit from the therapy, a non-typical LBBB (120–150 ms) showed rather detrimental therapy effects. The crucial point of interest was, that a rather narrow QRS complex will be widened up as the result of BiV pacing. Even though the QRS duration is not the best predictor for CRT response, it is still valuable to assess the success of resynchronization. Concluding from these findings, it became obvious that the BiV stimulation negatively affects the electromechanical sequence, as long as the initial VV dyssynchrony (<150 ms) is not pronounced enough. Besides the MADIT-CRT trial, several studies have been conducted and proven, that ideal patients for CRT, have an underlying LBBB. This leads to conduction through the RBB which is not affected and thus begin the ventricular activation in the RV. The LV stimulation occurs via the septum, which takes roughly 40–50 ms. In the presence of heart failure, this trans-septal conduction can be even more delayed. Another 50 ms are necessary to reach the posterolateral and ultimately 50 ms more to also activate the myocardium on the LV free wall. In total this leads to a QRS duration of ≥150 ms [16]. 

At this point Rickard et al. conducted a study observing the effect of CRT in 865 patients [17], 226 were already implanted with a RV pacemaker and then upgraded to compare their therapy response with the de novo cohort. Interestingly, this study did not exclude patients with previous chronic RV pacing (>85%). Furthermore, the established cut of point of >200 ms, was abandoned in this study as the baseline QRS duration was 187 ± 23 ms. Their results showed that independent of the initial QRS duration, RV introduced dyssynchrony and even people above the 200 ms cut off, equally profited from the CRT. Pre-CRT duration in the de novo group was 150 ± 26.1ms with an absolute change of 3.7 ± 28.4 ms and pre-CRT in the pacemaker-dependent patients at 187.8 ± 23.0 ms with an absolute change of −20.9 ± 36.3 ms. Vice versa their results leave some room for the observation, that the most pronounced CRT effect was also seen in those patients with LBBB > 150 ms and rather diminished around an LBBB with ≤150 ms. Furthermore, the Multisite stimulation in cardiomyopathy (MUSTIC) study included patients with an EF < 35%, NYHA III and QRS > 150 ms to receive a CRT [18]. Roughly half of the study group was in sinus rhythm, while the other half had a RV pacemaker during AF and were allowed to have a paced QRS duration of >200 ms. Despite a decrease in QRS width of 8–14% in the sinus rhythm and 14–24% in the AF group, they also observed moderate improvements in echocardiographic features. The LVEF in the sinus rhythm group improved from 22 ± 8 to 30 ± 2.1% and in the AF group from 26 ± 10 to 30.4 ± 7.8%. Although the results of the MUSTIC study show success at first glance, one still has to pay attention to the limitations of the BiV stimulation. Optimally placed electrodes could only restore the dyssynchrony of the ventricles and the electromechanical sequence to a limited extent. Looking at the results all together, the AF group in particular shows a small to perhaps moderate improvement. In synopsis of all the different approaches to improve CRT response, AF remains one of the biggest challenges yet to be overcome. 

As limited as the BiV pacing option might be in its versatility, the given option of an additional His bundle lead, is not only a good option for this special subgroup of CRT dependent AF patients. In a study group by Vijayaraman et al. patients with the same inclusion criteria as in the above mentioned studies, were implanted with an His-CRT-P/D. They showed remarkable results in resynchronization and normalization of echocardiographic parameters, even though their underlying diseases would have rendered the therapy with either sole His or only BiV as useless or suboptimal. Both approaches were tested alone and could only partially compensate the ventricular dyssynchrony, while combined they alleviated the conduction delay to a point, where one could speak about a narrow QRS complex. This promising novel approach has only been made use of in selected patients and has to prove its superiority over conventional CRT in larger randomized trials [9]. 

The newly published multi center MELOS study shows a good alternative to His bundle pacing in the form of left bundle branch area pacing. Either proximal, fascicular and left ventricular septal pacing were compared to each other with the consensus that a rather distal pacing strategy (LBFP, LVSP) would be more favorable [19]. Regarding the conduction system pacing, this might be a feasible alternative. Since our patients represent atypical LBBB morphology respectively IVCD, it can be assumed that the left bundle is in a desolate state which makes an individual approach in form of proximal–HBP and distal LV pacing necessary. In fact the LOT-CRT approach, coming from a similar group of researchers, proves that the HOT-CRT concept is working [20]. 

## 5. Limitations and Strengths

Our two patients do not make up for a large clinical trial to compare the effectiveness of HOT-CRT with the conventional approach. However, the pronounced improvements in either of our case reports and other respective studies, underline the importance of a closer look on this therapy option and substantiate its feasibility.

## 6. Conclusions

From our findings, we conclude that this approach seems to be a good option for patients suffering from AF, HF, and non-specific IVCD or non-LBBB that need a customized form of resynchronization. Moreover, it enhanced clinical and echocardiographic improvements in advanced HF patients with coexisting conduction disorders. 

Especially as the conventional CRT reaches its limits within this specific patient group, we need to think about alternative pacing sites and the most effective combination of them. 

## Figures and Tables

**Figure 1 medicina-58-01828-f001:**
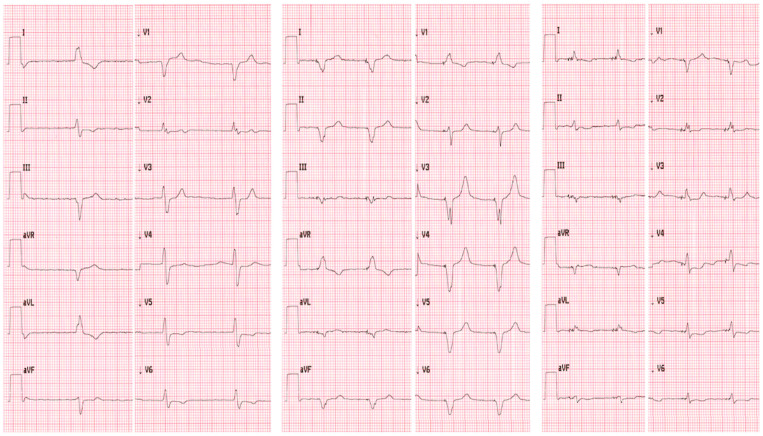
The 12-lead ECG of patient 1—initial QRS duration of 150 ms—on the left, BiV 140 ms—in the middle, HOT-CRT 120 ms—on the right.

**Figure 2 medicina-58-01828-f002:**
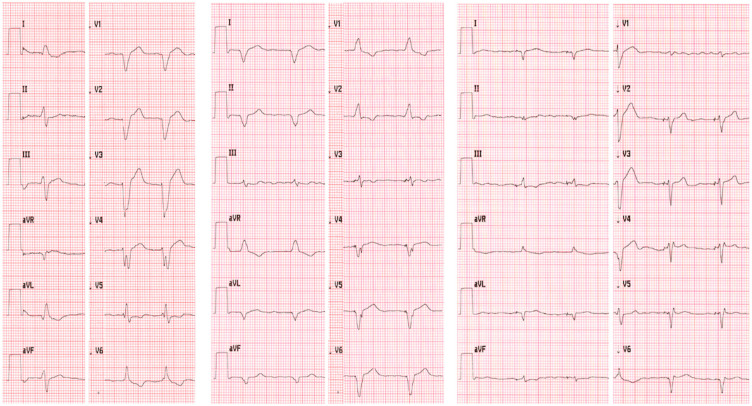
The 12-lead ECG of patient 2—initial QRS duration of 160 ms—left section, BiV 140 ms—middle section, HOT-CRT 120 ms—right section.

**Figure 3 medicina-58-01828-f003:**
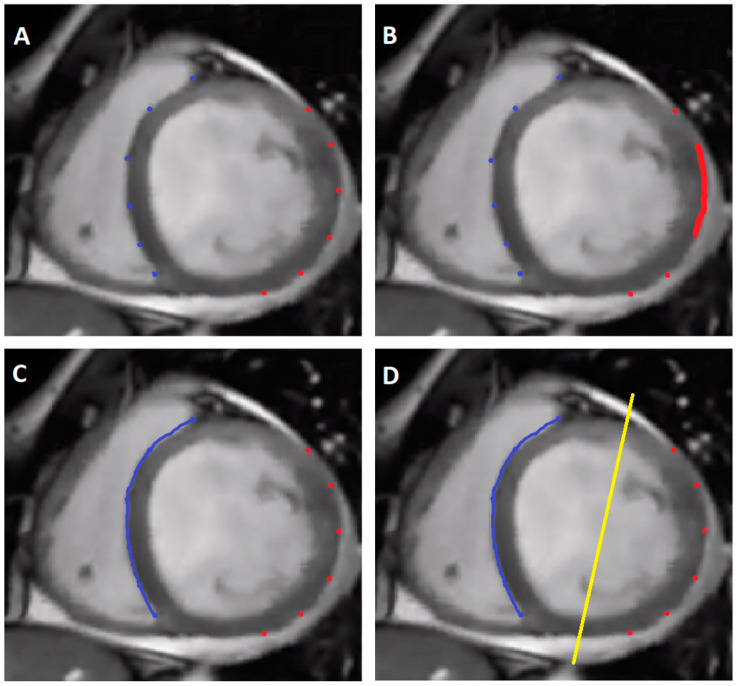
Schematic presentation of HOT-CRT concept vs. standard CRT. (**Panel** (**A**)): blue dots–RV pacing sites, red dots–LV pacing sites in CRT. To achieve optimal resynchronization the anatomic location of RV and LV leads should be exactly opposite which is not achievable for each patient. (**Panel** (**B**)): the red-marked area on the left ventricle indicates the most successful resynchronization LV electrode placement site. If the given patient does not have an appropriate target vein in this region the procedure result will not be optimal. (**Panel** (**C**)): In HOT-CRT concept the blue line represents RV activation area originating from His Bundle pacing via right bundle branch. (**Panel** (**D**)): Optimal resynchronization results from achieving perfect fusion of activation fronts coming from right and left site, as depicted by the yellow line. This effect is much more probable in HOT-CRT approach.

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
