# Peer review of "HOT CRT—The Effective Combination of Conventional Cardiac Resynchronization and His Bundle Pacing"

_medicina, 2022, doi:10.3390/medicina58121828_

Round 1

Reviewer 1 Report (Previous Reviewer 3)

Beside strain asynchrony can be assessed with other very well known methods as well, that are easily available on every echo machine.

His and parahisian (non-selective) pacing on current published data have similar outcomes. ECG recordings during implant or at follow-up should be easily available. 

The response provided in relation to information about LV EGM delay actually strongly suggest that in patient 2 what authors call HBP is actually not at all selective HBP (pacing from a HBP lead even without correction will not increase LV delay, or even shortens it if correction is present)

Authors added some explanations of figure 3 in the text but they did not improved at all figure 3 legend.

Author Response

Thank you very much for your review. On behalf of the authors, please find attached the answers to your suggestions.

Reviewer 2 Report (Previous Reviewer 2)

The authors answered the reviewer's questions to the best extent. Regarding previously raised question 4, the authors provided adequate answers. The reviewer would recommend adding the follow up pacing thresholds into the main text to improve the robustness of the study. 

Author Response

Thank you very much for your review. Please find attached the article with your suggested changes. 

This manuscript is a resubmission of an earlier submission. The following is a list of the peer review reports and author responses from that submission.

Round 1

Reviewer 1 Report

The authors presented two cases, demonstrating the efficacy of HOT-CRT in the setting of AF. The manuscript can be accepted for publication. However, the authors need to clarify the methods that they used to calculate the LVEF at baseline and during follow-up.

Reviewer 2 Report

Wolffe et al described 2 cases of His-bundle pacing optimzied CRT implants in patients with HFrEF, AF, non-typical LBBB with QRS>150ms and reported significant improvement in QRS duration, LVEF and NYHA functional class after 3-month follow up. The cases were described in detail and the discussions were comprehensive. However, some changes and additional explanations were recommended by the reviewer to improve the significance and presentation of the study. 

1) In page 3 line 81, what do the authors mean by "consecutive PCI LAD III“?

2) The authors reported the 2 patients in this study had chronic AF. Please specify if the patients had paroxysmal/persistent or permanent AF. What was the AF burden for each patient? During initial BiV pacing, what was the BiV pacing percentage? What rhythm control strategies were attempted on each patient?

3) What are some possible etiologies as to why LA size increased after HOT-CRT in the second patient?

4) Please report the lead threshold during implantation and during follow up as increasing pacing thresholds have been a concern for HBP. 

5) What was the author's definition for IVCD in these to patients? It would help clarify for readers if intracardiac electrograms can be shown to demonstrate the suspected location of the conduction block. 

6) Is figure 3 from the 2 patients reported? Please provide captions for each individual panel in Figure 3 in the figure legends in addition to the main text. 

7) In page 7 line 206, what is the "SN" group? Do the authors mean the sinus rhythm group?

8) Multiple grammatical error should be corrected in the text. 

Reviewer 3 Report

The authors presented 2 nice cases of General comments:

- some echocardiographic data on pre- and post implant LV asynchrony would be useful to prove the concept

- the type of selective His pacing (with or without correction) should be mentioned and at ECG tracings demonstrating it should be provided.

- the delay on LV lead electrogram should be provided (baseline as well as during selective His pacing)

Specific comments:

- in case 1 it should be mentioned that His lead was inserted to A lead port in the CRT-D device

- lines 121-122 should be changed from "Furthermore it can contribute LV pacing site to be less dependent on the venous anatomy" to " Furthermore His pacing will not change the area of LV maximum delay as clasic RV pacing might do"

- figure 3 should be better explained (for instance it is not clear what is the difference between A and B or C and D)

- the discussion on atrial fibrillation efect on CRT (lines 159 to 172) is wrongly directed as His pacing does not increase the % of efective pacing during AF (only AVN ablation can fix this issue). What is true (and worth to be mentioned) is that in HOT-CRT ventricular activation is not anymore related to an atrial event (as in fusion CRT, which depend on atrial event and therefore optimal fusion is lost during atrial fibrillation) 

- the discussion on RV pacing (lines 189-200) should be rather directed on benefits of His pacing vs RV in HF patients as well as in upgrade from RV pacing to His pacing (with/without additional LV pacing).